# Investigation of the Repairing Effect and Mechanism of Oral Degraded Sericin on Liver Injury in Type II Diabetic Rats

**DOI:** 10.3390/biom12030444

**Published:** 2022-03-13

**Authors:** Zhen-Zhen Wei, Yu-Jie Weng, Yu-Qing Zhang

**Affiliations:** School of Biology and Basic Medical Sciences, Medical College, Soochow University, RM702-2303, No. 199, Renai Road, Industrial Park, Suzhou 215123, China; 20204021006@stu.suda.edu.cn (Z.-Z.W.); 20184021002@stu.suda.edu.cn (Y.-J.W.)

**Keywords:** sericin, oral administration, protein expression, signaling pathway, inflammatory factors

## Abstract

In the sericulture and silk production industry, sericin is discharged in the degumming wastewater, resulting in a large amount of wasted natural protein and environmental pollution. This study investigated the effect of degraded sericin recovered by the Ca(OH)_2_–ultrasound degumming method (a green process) on liver injury in T2D rats. After 4 weeks of dietary sericin supplementation, the liver masses and organ coefficients of the T2D rats improved compared with those of the model rats that were not fed sericin. Oral sericin activated the damaged PI3K/AKT/AMPK pathway to enhance glycogen synthesis, accelerate glycolysis, and inhibit gluconeogenesis. The protein expression levels of the inflammatory factors NF-κB, IL-6, and TNF-α in the T2D model group were up to two times higher than in the normal group. However, all three T2D groups that received oral sericin showed significant decreases in these factors to the level found in the normal group, indicating that inflammation in the body was significantly reduced. These results show that the sericin protein might improve glycogen synthesis, accelerate glycolysis, and inhibit gluconeogenesis by enhancing the anti-oxidation capability and reducing inflammatory reactions. Therefore, sericin recovered by Ca(OH)_2_ degradation has potential use in the development of functional health foods that can lower blood sugar.

## 1. Introduction

The classic symptom of diabetes is insulin resistance, a primary cause of oxidative stress [1,2], which is reflected in increased levels of reactive oxygen species (ROS) in the liver. The liver is the most important target organ of insulin [3]. In the liver, insulin binds to the insulin receptor on the cell membrane to activate the insulin receptor substrate, which in turn binds to the regulatory subunit p85 of PI3K (phosphatidylinositide3-kinases) and activates catalytic subunit p110 of PI3K. Activated PI3K generates a second messenger and promotes the activation of AKT (protein kinase B). Therefore, the insulin/PI3K/AKT pathway plays an important role in regulating liver carbohydrate metabolism. Studies have found that it can be an important part of the treatment of diabetes, cancer, neurodegenerative diseases, and stroke targeting; AMPK (AMP-dependent protein kinase) is an important sensor molecule that regulates bioenergy homeostasis, which can control the synthesis and decomposition pathways of energy metabolism. AMPK activation can regulate the metabolism and vascular function of adipocytes, promoting glucose transport and fatty acid oxidation. The liver is also the main organ that regulates blood sugar levels. When blood sugar levels rise after meals, the liver uses blood sugar to synthesize glycogen. Excessive sugar is converted into fat in the liver and accelerates the pentose phosphate cycle, thereby reducing blood sugar and maintaining a constant blood sugar concentration. Conversely, when the blood sugar concentration decreases, the hepatic glycogenolysis and gluconeogenesis are strengthened, and the generated glucose is sent into the blood to adjust the blood sugar concentration so as not to be too low.

Metformin is one of the most commonly used oral drugs clinically used to lower blood glucose levels in diabetic patients. After issuing warnings for decades about the risks of lactic acidosis in patients with chronic nephropathy, most evidence from the recent literature has demonstrated that the risk of lactic acidosis is low and acceptable [4]. Acarbose can cause gastrointestinal reactions [5]. Thiazolidinediones may cause water and sodium retention, bone looseness, etc. [6]. Therefore, natural medicines with a reliable anti-hyperglycemic effect that are less toxic and have fewer side effects have received increasing attention [7]. Silk sericin protein from *Bombyx mori* cocoons contains 18 essential amino acids. Sericin is biologically active [8] and has strong physiological and pharmacological effects such as anti-oxidation [9,10], antitumor [11,12,13], anti-apoptosis [14], and blood sugar reducing activities [15,16]. Sericin has also been shown to reduce serum cholesterol in rats [17,18] and enhance cognition in Alzheimer’s disease patients [19,20]. Our group has focused on the green processing and recovery of sericin from silkworm cocoons and its anti-hyperglycemic effects, shown in vitro and in vivo. It has recently been reported that sericin added to the diet can effectively reduce blood sugar levels in a streptozotocin-induced type II diabetic (T2D) mouse model. It has also been found to reduce intestinal cholesterol, triglycerides, and other lipids and improve glucose and insulin tolerance in mice [21].

We recently published the results of the first half of the present study, which seeks to better understand the blood-sugar-lowering function of sericin and its regulatory mechanism. Degraded sericin was shown to regulate blood glucose levels and improve impaired liver function in T2D rats by reducing oxidative stress [22]. At the same time, some studies have also found that mulberry leaf polysaccharides can reduce inflammatory factors in T2D rats, improve mitochondrial function and relieve oxidative stress damage, and have an anti-diabetic effect [23]. The present article reports the second half of the study, which investigated the effect of sericin administration on T2D rats. We explored the ability of sericin to repair liver damage and its mechanism of action in diabetic rats in terms of glucose and lipid metabolism, the PI3K/AKT signaling pathway, and the AMPK/ACC signaling pathway.

## 2. Materials and Methods

### 2.1. Materials

Western blot antibody, TNF-α (tumor necrosis factor-α), IL-6 (interleukin- 6), and NF-κB (nuclear factor kappa-B) enzyme-linked immunosorbent assay (ELISA) kits, Jiangsu KGI Biotechnology Co., Ltd., Nanjingn China; bicinchoninic acid assay (BCA) determination kit, Shanghai Biyuntian Institute of Biotechnology, Shanghai, China; multifunctional sample homogenizer, QIAGEN Biotech, Hilden, Germany; MIKro120 low-speed centrifuge, Germany Hettich Scientific Instruments Co., Ltd., Kirchlengern, Germany; desktop high-speed refrigerated microcentrifuge, SCILOGEX, Czech Republic, USA. The rest of the reagents are analytical pure.

### 2.2. Preparation of Sericin Peptides

We used an improved degumming method developed by our laboratory to obtain the sericin peptides [24]. First, the silk underwent degumming by boiling in 0.025% (*w*/*v*) calcium hydroxide for 30 min; this boiling step was repeated once to increase the recovery rate. This was followed by rotary evaporation of the obtained degumming liquid to concentrate it. The concentrated solution was neutralized with 6M dilute sulfuric acid and centrifuged at 10,000 rpm/min to remove the precipitate. Finally, the supernatant underwent vacuum freeze-drying to obtain the low-molecular-weight sericin (referred to hereafter as LS). In this study, LS samples containing 1%, 2.5%, and 5% were defined as LLS, MLS, and HLS, respectively, for the subsequent feeding of T2D rats.

### 2.3. Rat Feed

Sprague–Dawley (SD) rats are often used as a diabetic animal model due to their higher sensitivity to streptozotocin (STZ) and their easily attainable and inexpensive characteristics. This can help further elucidation of the underlying mechanisms of diabetes and its complications [25,26,27,28]. The procedures for the rat breeding and handling, model construction, blood glucose measurement, and anatomical measurement in the experiment were carried out according to our previous report [22]. Purchased male SD rats were raised to approximately 200 g in a controlled environment at 20–25 °C and 50%–80% humidity. In this study, rats fed with metformin were used as the positive control group. We randomly tested several STZ injection doses for T2D modeling, and the 90 mg/kg dose was identified as the optimal dose for use in the study. The rats fasted overnight before the first STZ dose, and they received one dose per day for 3 days. Two days after the final injection, blood was collected from the tail vein on an empty stomach to test the fasting blood glucose level. The model successfully established if the fasting blood glucose was higher than 11.1 mmol/L. The rats were randomly divided into six groups (*n = 6*). The normal group and T2D model group were fed normal irradiated feed. Three sample groups of T2D rats were fed irradiated chow containing LLS, MLS, and HLS samples. The positive control (PC) group was fed irradiated feed containing 0.5% (*w*/*w*) metformin. The entire test process was completed in 7 weeks. All animal experiments were conducted in accordance with the relevant regulations required by the International Animal Welfare Committee and the Animal Experiment Operation and Ethics Committee of Soochow University.

### 2.4. Liver Weight and Organ Body Coefficients

After the rats were sacrificed by dislocation, the liver was removed and weighed. The organ coefficient was calculated as liver mass/rat weight × 100%.

### 2.5. Extraction of Total Protein from Liver Samples

First, 10 μL of phosphatase inhibitor, 1 μL of protease inhibitor, and 5 μL of 100 mmol/mL PMSF were added to 1 mL of prechilled lysis buffer and mixed well. The samples were kept on ice for a few minutes until use. Each 100 mg liver was cut into 3 mm pieces. One milliliter of chilled lysis buffer containing inhibitors was added to each 3 mm piece, and the sample was homogenized at 50 r/min. All operations were carried out at low temperature. After homogenization, the sample was centrifuged for 5 min at 10,000 r/min and 4 °C. The supernatant containing the whole protein extract was divided into two precooled centrifuge tubes and stored at −70 °C to avoid repeated freezing and thawing. The protein concentration of the obtained supernatant was determined using a BCA protein concentration kit.

### 2.6. Western Blotting

Total protein content in liver tissue was extracted and separated by SDS-PAGE electrophoresis. The separated proteins were transferred to NC membranes. Protein-transferred membranes were then blocked with 5% nonfat milk for 2 h at room temperature and incubated with dilution-related antibodies overnight at 4 °C. Membranes were washed and then mixed with Rabbit Anti-AKT (1:10,000), Rabbit Anti-p -ACC (1:5000), Rabbit Anti-P-ULK1 (1:500), and other corresponding antibodies diluted 1:1000 and incubated at room temperature for 1.5 h. After imaged using G:BOX chemiXR5., the results were analyzed in grayscale using Gel-Pro32 software.

### 2.7. Determination of Related Inflammatory Factors

According to the instructions of each ELISA kit, the contents of TNF-α, IL-6, and NF-κB in the supernatant of the liver homogenate of each group of rats were detected.

### 2.8. Data Processing

The data obtained from the experiment were processed using Prism 8 statistical software. The results are presented as means ± standard deviation. One-way analysis of variance was conducted, and *p* < 0.05 indicates statistical significance.

## 3. Results

### 3.1. Liver Organ Quality and Organ Coefficient

The liver is the most important organ in nutrient metabolism. Diabetes can disrupt fat metabolism, affect liver function, and cause organ enlargement. The liver masses and organ coefficients of the rat groups were compared, and the results are shown in Table 1. The normal group had the largest average liver mass (16.38 ± 1.70 g). The liver mass of rats in the T2D model group was greatly reduced (10.86 ± 1.43 g; *p* < 0.01). The liver quality of the PC group was significantly increased compared with the T2D model group (*p* < 0.01) and was very close to that of the normal group. The liver mass of the HLS group was also increased to a certain extent, but the other sericin test groups did not have significantly different liver masses. The organ coefficient refers to the ratio of the organ to the body weight. If the coefficient increases, the organ may show signs of hyperemia, edema, or hyperplasia. The organ coefficient of rats in the normal group was 3.25 ± 0.34%, while that of rats in the model group was 4.34 ± 0.54%, which was significantly greater (*p* < 0.01). The organ coefficients of the PC group and each sericin group were increased compared with the normal group, but the increases were not significant. The coefficient of the HLS group was significantly decreased compared with the model group (*p* < 0.05). These results indicate that HLS can improve liver hyperplasia and hypertrophy caused by diabetes.

### 3.2. Carbohydrate Metabolism Pathway

The unbalanced blood glucose levels in T2D rats were largely due to disordered gluconeogenesis and glycolysis metabolism. In a high-glucose environment, Glucose-6-phosphatase (G6pase), phosphoenolpyruvate carboxylase (PCK), and glucokinase (GLK) are involved in the conversion and storage of glucose. Glycolysis is the process in which sugars are broken down to produce lactic acid and energy. The main rate-limiting enzymes of glycolysis are 6-phosphofructokinase-1 (PFK1) and M2 pyruvate kinase (PKM2). This study investigated these key enzymes of sugar metabolism. As can be seen from Figure 1, the levels of G6pase and PCK in the model group and PC group were higher than those in the normal group, and the protein level in the HLS group basically recovered to the level of the normal group, but the expression levels of G6pase in the LLS and MLS groups were not significantly up-regulated, indicating that HLS-fed rats can accelerate the conversion and storage of glucose; the expression levels of GLK, PFK1, and PKM2 in each experimental group were higher than those in the normal group and the model group, and the indicators in the HLS group were close to or even higher than those in the PC group. Compared with the PC group, the expression levels of LLS and MLS were also increased, indicating that feeding LS could accelerate the glucose conversion and weaken the glycolytic activity. In general, these results indicate that feeding with a higher amount of sericin can evidently reduce the blood sugar levels by promoting gluconeogenesis and weakening glycolysis.

### 3.3. Impact on PI3K/AKT Pathway

Insulin is the main regulatory hormone in the process of glucose and lipid metabolism, and insulin mainly regulates glucose uptake and transport through the PI3K/AKT pathway. PI3K, AKT, fructose-2,6-bisphosphatase cardiac enzyme (PFKFβ2) and forkhead box protein O1(FoxO1) in the liver are the key regulatory enzymes. After PI3K activation, phosphate groups can be recruited to the plasma membrane of AKT protein to partially phosphorylate and activate, and p-AKT will further activate the downstream PFKFß2 and FOXO1. As shown in Figure 2, compared with the model group, with the increase in the sericin concentration, PI3K, p-PI3K, p-AKT, PFKFß2, p-PFKFß2, and p-FOXO1 in the three dose groups were gradually increased, while the indexes in the HLS group almost returned to normal levels; FoxO1 in the sericin treatment group decreased, and the HLS group decreased by about 50%. Compared with the PC group, the levels of p-PI3K, p-AKT, PFKFß2, p-PFKFß2, and p-FOXO1 in the three dose groups were gradually increased, and the HLS group basically reached the same level as the normal group; for PI3K and AKT in the MLS group, the protein content was almost the same as that of the PC group, which exceeded the level of the normal group; the FOXO1 protein level of the HLS group was almost the same as that of the PC group, and it was reduced to the level of the normal group; for PI3K, p-PI3K, and p-AKT, the levels of PFKFß2 and p-FOXO1 were significantly lower than those in the normal group, there was almost no difference in the amount of total AKT and p-PFKFß2, the level of FoxO1 was significantly increased, and various phosphorylated proteins were highly expressed in the HLS group, indicating that the HLS group that activated the PI3K/AKT pathway has the strongest effect. The above results indicate that adding sericin can inhibit hepatic gluconeogenesis and sugar production and increase the synthesis and storage of hepatic glycogen, thereby stabilizing the blood sugar level of the body.

### 3.4. Impact on AMPK/ACC Pathway

AMPK is a key protein in energy metabolism and is of great significance to the study of metabolic diseases in diabetes. Glucose transporter 4 (GLUT4) exists only in insulin-sensitive tissues such as hepatocyte membranes and has a high affinity for glucose, so it is involved in insulin-stimulated glucose transport. Although GLUT2 has a high specificity for glucose, it has a high affinity for glucosamine, so this study investigated the expression of GLUT4 in the liver of rats in each group. GLUT4 accelerates cellular uptake of glucose, which in turn accelerates hepatic glycogen synthesis, ultimately lowering blood glucose levels. Excessive blood glucose concentrations stimulate insulin secretion, resulting in a dramatic increase in GLUT4 expression on the membrane, accelerating glucose conversion and storage [29]. When the intracellular AMP/ATP ratio increases, activated AMPK can enhance glucose uptake, improve insulin sensitivity, promote fatty acid oxidation, and reduce sugar and lipid production. Liver kinase B1 (LKB1) is the major upstream kinase of AMPK and is closely related to intracellular energy metabolism, muscle contraction, and ATP regeneration. Acetyl-CoA carboxylase (ACC) is an important substrate of AMPK and plays an important role in regulating fatty acid synthesis, which can not only inhibit the production of cholesterol and fat, but also enhance fatty acid oxidation. In Figure 3, the LKB1, p-LKB1, p-AMPK, p-GLUT4, and GLUT4 in the model group were significantly lower than those in the normal group (*p* < 0.01). The levels of AMPK and ACC in the model group were higher than those in the normal group (*p* < 0.05). The expression levels of the three experimental groups showed a downward trend, which was similar to the PC group but did not reach the level of the normal group, indicating that AMPK/ACC was activated in T2D rats after feeding with sericin; the expressions of p-AMPK, p-LKB1, p-GLUT4, GLUT4, ACC, and p-ACC were up-regulated. It almost recovered to the level of the PC group and even partially reached the level of the normal group, in which the high expression of GLUT4 can more directly show that the glucose level in T2D rats is accelerating. Increased enzyme expression levels accelerate glycolysis; activated AMPK can also inhibit ACC production, thereby inhibiting fat synthesis.

### 3.5. Impact on Liver Damage

Oxidative stress in the liver of T2D mice triggers a cascade of reactions, resulting in inflammation. NF-κB is an important immune-related transcription factor that is involved in the transcriptional regulation of a variety of cytokines. A large amount of IL-6 in a high-glucose environment can reduce insulin sensitivity [30], and excessive TNF-α will cause an increase in the free fatty acid content, which will eventually lead to insulin resistance [31]. Figure 4 shows that the levels of NF-κB, IL-6, and TNF-α in the liver of the T2D model group were significantly higher than those of the normal group (*p* < 0.01) by 2.1-fold, 1.3-fold, and 1.2-fold, respectively. These results show that severe inflammation occurred in the livers of the diabetic rats. After 4 weeks of feeding with sericin, the NF-κB, IL-6, and TNF-α levels in the three dose groups were all downregulated to normal levels in a dose-dependent manner (*p* < 0.05, *p* < 0.01). In all three oral sericin groups, the anti-TNF inflammatory factor was strong and almost the same as in the PC group. These results show that adding sericin effectively reduces the inflammatory response in the livers of diabetic rats.

## 4. Discussion

Typical symptoms of diabetes include insulin resistance (IR) and accumulation of triglycerides in the liver, which not only aggravate the progression of diabetes, but may also lead to nonalcoholic fatty liver disease (NAFLD). NAFLD affects more than 20% of adults [32,33]. NAFLD also has the potential to develop into diseases such as cirrhosis, liver failure or hepatocellular carcinoma. The liver is the most critical target organ of insulin [9,34], and insulin is the only hormone that can lower blood sugar. Insulin mainly regulates glucose absorption, glycogen synthesis, and degradation through insulin regulation through the PI3K/AKT signaling pathway [16,35]. In the liver, insulin binds to the insulin receptor IR on the cell membrane, then activates the IR substrate (IRS) and then activates PI3K to generate a second messenger that promotes the activation of AKT. p-AKT regulates the carbohydrate metabolism process of hepatocytes through the following pathways. First, AKT activation promotes the transfer of GLUT4 to the cell membrane and promotes the inward transport of extracellular glucose [36]. Second, p-AKT mediates glycogen-forming enzyme kinase 3 (GSK3) [37]. Third, p-AKT inhibits hepatocyte gluconeogenesis through peroxisome receptor-γ coactivator-α [38]. Therefore, the insulin-PI3K/AKT pathway plays an important role in regulating the hepatic carbohydrate metabolism. Studies have shown that adding sericin to the daily diet can not only effectively reduce the blood sugar level in the STZ-induced T2D rat model [39,40], but also reduce the intestinal cholesterol, triglyceride, and other lipid intakes while improving the body’s tolerance to glucose [41]. AMPK is an energy sensor in cells. Activated AMPK can promote GLUT4 transport and increase glucose uptake [42]. Sericin supplementation resulted in up-regulation of LKB1, AMPK, and GLUT4 and down-regulation of ACC protein expression. This suggests that sericin may accelerate glycolysis and reduce the body’s absorption of fat by activating AMPK.

Sericin is a glue that binds silk fibroin fibers together to strengthen silkworm cocoons. In the industrial processing of raw silk, it is often discharged together with degumming agent as alkaline waste, which not only wastes biological resources, but also pollutes the environment. Therefore, its green recycling and efficient utilization has always been the focus of research and development in the silk industry [43,44]. The utilization of sericin mainly involves two aspects: industrial materials and biological actives. The larger molecular weight is suitable for the former application [8], while degraded sericin and even hydrolysate are suitable for the latter application [45]. In the past ten years, our team has been engaged in the green recovery of sericin, the preparation of degraded sericin, and its application in hypoglycemic functional foods, etc. [46,47]. Recently, our group newly developed a green technology for degumming silk fiber with Ca(OH)_2_ aqueous solution [24]. It was found that this degumming method can recover the low molecular weight of degraded sericin peptide and its hydrolysate, which not only have strong antioxidant activity, but also inhibition activity on glycosidase and tyrosinase in vitro [48]. Colleagues from Chengde Medical University (Chengde, Hebei 067000, P.R. China) also investigated the hypoglycemic effect of sericin water extract on rats and its mechanism. The sericin water extract could enhance the insulin-PI3K/AKT signaling pathway in the liver of a rat model of T2D [49]. It also reduced hippocampal neuronal apoptosis in a model rat by activating the AKT signal transduction pathway [50] and has a protective effect on the nervous system in diabetic rats [33,51]. However, it should be noted here that the sericin sample used in that experiment was an aqueous extract of sericin obtained by boiling water from the shell layer of a colored silkworm cocoon, which contains a lot of β-carotene with anti-oxidation. The above anti-diabetic effects are probably the result of the combined action of high-molecular-weight sericin and β-carotene. In our recently published paper [22], orally administered sericin samples were degraded sericin peptides and their hydrolysates obtained by the Ca(OH)_2_ method from the shells of white silkworm cocoons that are now commercially available. After the oral administration of sericin to diabetic model rats, the pathological slices showed that the rat liver cells in the model group were severely damaged, with inflammatory infiltration and edema. Furthermore, the serum levels of ALT, AST, and ALP were significantly increased, which also indicated liver damage. These results show that adding sericin to the diet alleviated diabetic liver damage. The present paper is an in-depth investigation of the aforementioned research.

As shown in Figure 5, after oral administration of sericin, the first noticeable effect was an increase in the antioxidant capacity of the T2D rats, followed by a significant decrease in the level of inflammatory factors in the body. This decrease resulted in changes in the expression of genes and proteins related to insulin synthesis and secretion, glucose synthesis and decomposition, glycogen synthesis and storage, and even the synthesis and metabolism of fat. These changes might ultimately lead to a gradual improvement in insulin resistance in diabetic rats and the normalization of blood sugar levels. The present results are mostly consistent with our previous results obtained with T2D mice. In the future, we will continue to investigate orally administered sericin and the ability of functional foods to lower blood sugar. This new use of sericin provides a reason for utilizing this waste product of silk processing and will support the sustainable development of the silk industry.

## 5. Conclusions

Sericin is a natural macromolecular protein with various biological activities, such as antioxidation and hypoglycemia. In this paper, the hypoglycemic mechanism of low-molecular-weight sericin (LS) prepared by the ultrasonic degumming method and Ca(OH)_2_-ultrasonic degumming method after feeding T2D rats was investigated. The results show that all the concentrations of LS can improve the liver hyperplasia and hypertrophy caused by diabetes; at the same time, it was found that different concentrations of LS can regulate G6pase, GLK, and PCK related to the glucose metabolism pathway, and regulate the PI3K/AKT and AMPK/ACC pathways to enhance glycogen synthesis, accelerate glycolysis, and inhibit gluconeogenesis. It can also reduce the expression of inflammatory factors TNF-α, IL-6, and NF-κB, and significantly relieve liver oxidative stress.

## Figures and Tables

**Figure 1 biomolecules-12-00444-f001:**
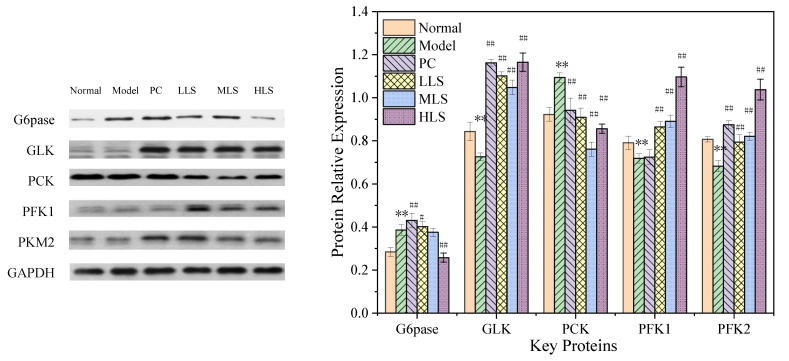
Effect of oral sericin on the expression of key hepatic glucose metabolism proteins in T2D rats. Normal: normal group; Model: diabetic model group; PC: 0.5% melbin; LLS, MLS, and HLS: 1%, 2.5%, and 5% LS, respectively. *n* = 3, ** *p* < 0.01 versus normal group. ^#^ *p* < 0.05, ^##^ *p* < 0.01 versus model group.

**Figure 2 biomolecules-12-00444-f002:**
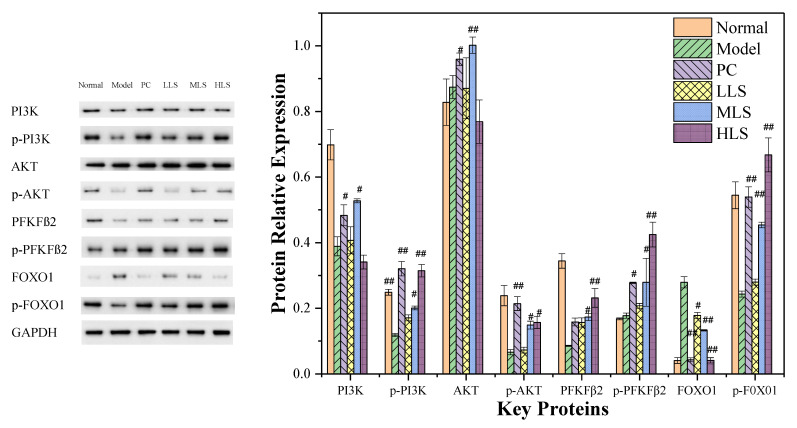
Effect of oral sericin on the expression of PI3K/AKT pathway proteins in T2D rats. Normal: normal group; Model: diabetic model group; PC: 0.5% melbin; LLS, MLS, and HLS: 1%, 2.5%, and 5% LS. *n* = 3, ^#^ *p* < 0.05, ^##^ *p* < 0.01 versus model group.

**Figure 3 biomolecules-12-00444-f003:**
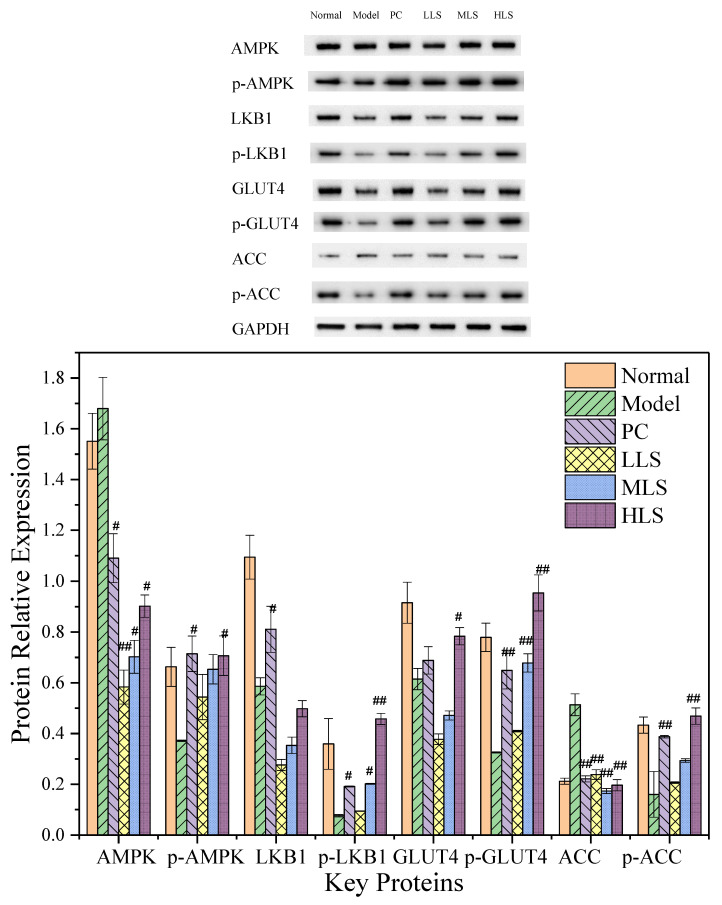
Effect of sericin on the expression of key AMPK/ACC proteins in T2D rats. Normal: normal group; Model: diabetic model group; PC: 0.5% melbin; LLS, MLS, and HLS: 1%, 2.5%, and 5% LS. *n* = 3, ^#^ *p* < 0.05, ^##^ *p* < 0.01 versus model group.

**Figure 4 biomolecules-12-00444-f004:**
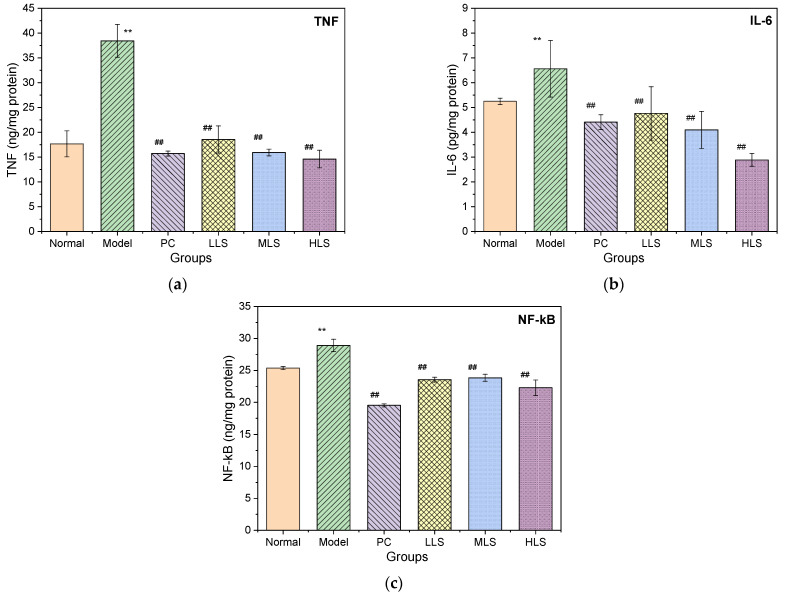
Effect of oral sericin on TNF-α, IL-6, and NF-κB levels in livers of T2D rats. (**a**–**c**) are TNF-α, IL-6, and NF-κB levels, respectively. Normal: normal group; Model: diabetic model group; PC: 0.5% melbin administration; LLS, MLS, and HLS: 1%, 2.5%, and 5% L. *n* = 3, ** *p* < 0.01 versus normal group. ^##^ *p* < 0.01 versus model group.

**Figure 5 biomolecules-12-00444-f005:**
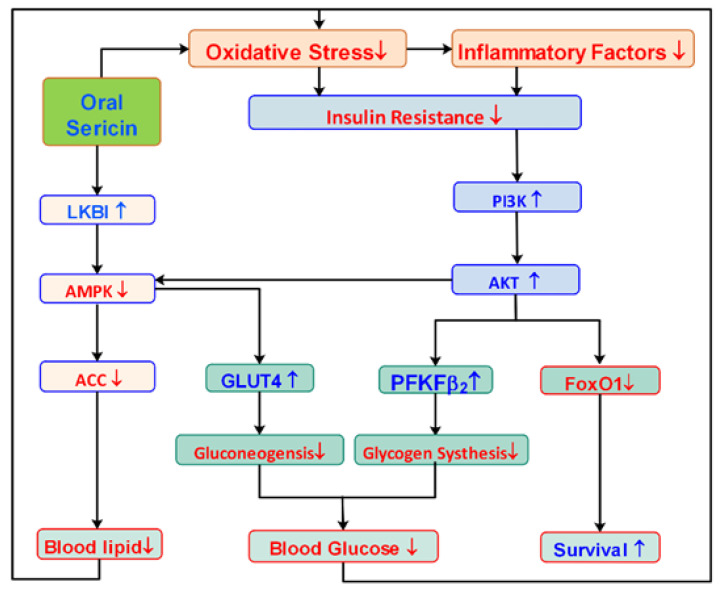
Possible pathways through which oral sericin might affect T2D rats. ↓ & ↑, means upstream and downstream of the pathway and feedback regulation, respectively.

**Table 1 biomolecules-12-00444-t001:** Effect of LS oral administration on liver mass and liver coefficient of T2D rats.

Group	Liver Mass (g)	Coefficient (%)
Normal	16.38 ± 1.70	3.25 ± 0.34
Model	10.86 ± 1.43 **	4.35 ± 0.54 **
PC	15.73 ± 2.92 ^##^	4.23 ± 0.79
LLS	10.43 ± 1.20	4.11 ± 0.49
MLS	10.49 ± 1.33	4.23 ± 0.54
HLS	11.90 ± 1.34 ^#^	4.03 ± 0.73 ^#^

Normal: normal group; Model: diabetic model group; PC: positive control group (0.5% metformin); LLS, MLS, and HLS: 1%, 2.5%, or 5% LS, respectively. *n* = 3, ** *p* < 0.01 versus normal group. ^#^ *p* < 0.05, ^##^ *p* < 0.01 versus model group.

## Data Availability

Code and material; The datasets used and/or analyzed during the current study as well as analysis scripts are available from the corresponding author on reasonable request.

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
