# Peer review of "Investigation of the Repairing Effect and Mechanism of Oral Degraded Sericin on Liver Injury in Type II Diabetic Rats"

_biomolecules, 2022, doi:10.3390/biom12030444_

Round 1

Reviewer 1 Report

The work entitled "Repairing effect and mechanism of oral degraded sericin on impact of disease 2 in type II diabetes mellitus" contains potentially relevant data for the implementation of alternative type 2 diabetes pharmacology algorithms and for the treatment of overweight and obesity using naturally derived active substances. The authors of this publication describe the effect of degraded sericin on the liver tissue of diabetic rats.

The authors of this publication suggest that sericin could potentially be used as a supplement to regulate blood glucose levels.

Comments: 

1.The author's statement [Metformin is one of the most commonly used oral drugs clinically used to lower blood glucose 38 levels in diabetic patients, and its long-term use may cause lactic acidosis; Introduction section] based on one cited article is highly misleading. The truth is that the risk of lactic acidosis is low or even extremely rare.

  "... the reported incidence of lactic acidosis in clinical practice has proved to be very low (<10 cases per 100,000 patient-years) ...." [RalphDeFronzoaG. AlexanderFlemingbKimChencThomas A. Bicsakc; Metformin-associated lactic acidosis: Current perspectives on causes and risk; Metabolism; Volume 65, Issue 2, February 2016, Pages 20-29.]

„…Metformin is currently considered a first-line therapy in type 2 diabetic patients. After issuing warnings for decades about the risks of lactic acidosis in patients with chronic nephropathy, metformin is now being re-evaluated. The most recent evidence from the literature has demonstrated both a low, acceptable risk of lactic acidosis and a series of favorable effects, which go beyond its hypoglycemic activity.…”

Filippo Mariano, Luigi Biancone; Metformin, chronic nephropathy and lactic acidosis: a multi-faceted issue for the nephrologist; J Nephrol. 2021 Aug;34(4):1127-1135. doi: 10.1007/s40620-020-00941-8. Epub 2020 Dec 29.

The method section does not include information on advantages and disadvantages of the used in the study the animal rat model.

The method section does not include information on standarisation of the animal rat model.

The title of 2.5. section ”Extraction of whole protein from liver samples” should be changed – i suggest ”Extraction of total protein from liver samples”

In the Discussion section, the Authors of the manuscript cite their own paper, but in fact, in this chapter, the Authors should not only refer to the results/conclusions of comparable (already published) experiment, but first of all the Authors of this manuscript sholud propose an attempt of the explaination the obtained results, described in this paper.

Therefore, in its present form, the Discussion section representing a summary of the obtained results – strongly needs to be remodeled.

The Conclusion chapter is not a direct and convincing answer to the purpose/aim of the work - the Conclusion chapter should be remodelled.

Finally, the title ("Repair effect and mechanism of action of orally degraded sericin on liver injury 2 in rats with type II diabetes") of the peer-reviewed article "Repair effect" was not convincingly proven / described in the text the manuscript in question. 

Reviewer 2 Report

The current paper by Zhen-Zhen Wei and colleagues reports on the repairing effect and mechanism of oral degraded sericin on liver injury 1 in type II diabetic rats. Although the information being reported could be potentially useful, the paper is still flawed and contains many errors that make it misleading at the end.

Specific comments:

  1. The introduction still contains many statements without accurate citations.
  2. Abbreviations are not all described at first mention. Some abbreviations are not consistent, and make the paper difficult to follow.
  3. There are many molecular mechanisms being discussed (reported) but without a strong motivation for their exploration in the first place.
  4. Materials are only reported for only a few parameters (like inflammatory markers) and not for all assays performed. This is essential, especially giving catalogue numbers of antibodies, for repetition of experiments.
  5. Motivate for selection and duration of all treatments, including use of positive controls
  6. Western blot is not described clearly
  7. Statistical analysis is not described clearly
  8. Figure captions are shallow
  9. Some Western blots are not clear, while others do not show any difference between control and model
  10. How do you explain the expression of GLUT4 in the liver?
  11. Manuscript needs thorough reading to eliminate errors, also not using a mixture of small and capital letters without a reason
  12. Most of the information within the discussion is irrelevant, this section should be re-written

Round 2

Reviewer 1 Report

The Authors of the publication "Repairing effect and mechanism of oral degraded sericin on liver injury in type II diabetic rats" submitted to the editorial board of the Biomolecules Journal properly addressed the reviwer's remarks/suggesntions - the manuscript has been efficiently corrected.
Therefore I recommend to publish the modified article - without any corrections - in its present/current form. 

Author Response

Thank you very much for your valuable comments on this article.

Reviewer 2 Report

The paper has been slightly improved, and it should be considered for publication, especially considering that its an add-on to already published results, from the same group. However, my outstanding major concern is about GLUT4 expression (detected by WB) in the liver. Please note, GLUT4 expression (even at mRNA level) in liver is extremely very low. Could the cross contamination of tissue sample affected the results? Provide strong motivation for detection of GLUT4 protein expression on the liver or rather remove this part (throughout the paper), to solidify the presented evidence.

Author Response

After receiving your comments last time, we carried out the relevant phosphorylation experiments and repeated the GLUT4 protein expression experiments, but the GLUT4 map was not updated together when the revised manuscript was uploaded. The changes are reflected in Figure 3. Thanks to the reviewer for the reminder.

Round 3

Reviewer 2 Report

The authors still don't justify the detection/expression of GLUT4 in the liver, especially knowing that GLUT2 is the major isoform found in the liver. A conversation about this should be added within the manuscript.

Author Response

Thank you very much for your suggestion. The English language ,style and relevant content has been modified and explained within the manuscript.

Have add following paragraph in manuscript:

Glucose transporter 4 (GLUT4) exists only in insulin-sensitive tissues such as hepatocyte membranes and has a high affinity for glucose, so it is involved in insulin-stimulated glucose transport. Although GLUT2 has a high specificity for glucose, it has a high affinity for glucosamine, so this study investigated the expression of GLUT4 in the liver of rats in each group. GLUT4 accelerates cellular uptake of glucose, which in turn accelerates hepatic glycogen synthesis, ultimately lowering blood glucose levels. Excessive blood glucose concentrations stimulate insulin secretion, resulting in a dramatic increase in GLUT4 expression on the membrane, accelerating glucose conversion and storage.

This manuscript is a resubmission of an earlier submission. The following is a list of the peer review reports and author responses from that submission.

Round 1

Reviewer 1 Report

This is a continuous project on the therapeutic properties of sericin in a type 2 diabetes rat model. Although of potential interest, the manuscript lacks essential information that is necessary to build on initially published evidence, to determine the potential therapeutic mechanisms implicated in the protective effects of Sericin against liver injury in a state of type 2 diabetes.

Specific comments:

Authors fail to highlight the significance of investigating the molecular mechanisms PI3K/AKT or AMPK/ACC (which is basically insulin-dependent/independent regulation)

The manuscript has many statements/sections without accurate citations

For the repetition of experiments, authors should provide all sources of materials and reagents

What were the sources of antibodies? The indicated dilution for Western blot, does it apply for all antibodies?

Did the authors provide the methodology for the quantification of inflammation assays?

The authors should look/provide data for phosphorylation of AKT/PI3K/AMPK, to solidify presented evidence

The discussion is defining terms instead of highlighting the significance of results, especially in relation to published literature…

Abbreviations are not defined

Figure captions are not informative

Reviewer 2 Report

The paper entitled “Repairing effect and mechanism of oral degraded sericin on liver injury in type II diabetic rats” includes potentially relevant data for the hepatological injury repair precisely focused of the pathologically changed diabetic repair process. The Authors of the manuscript observed that the sericin molecule may be responsible for the glycogen synthesis enhancement, glycolysis accelerating, and gluconeogenesis reduction by enhancing the anti-oxidation capability and modulating the inflammatory response. Moreover the Authors tend to conclude that recovering of the sericin by sericin recovered by Ca(OH)2 degradation possesses significant potential acting also as a basis for the design and implementation of professional functional health foods products directed toward the enhancement of the impaired healing during disturbed glucose metabolism.

However, I have a few remarks:

  1. The sixth sentence of Introduction section “Metformin is a clinically available drug for treating diabetes.”, strongly needs to be remodeled. As far as I know drugs (considered as an active pharmaceutical substances) in the course of diabetes – and not only diabetes – are used to treat the patient rather than the disease.
  2. Referring to the current state of knowledge: “Metformin: decreases hepatic glucose output by inhibiting gluconeogenesis; increases insulin-mediated glucose utilization in peripheral tissues (such as muscle and liver), possesses antilipolytic effect that lowers serum free fatty acid concentrations, thereby reducing substrate availability for gluconeogenesis [Diabetes. 1987 May;36(5):632-40.; J Clin Endocrinol Metab. 1991 Dec;73(6):1294-301.; Diabetologia. 2017;60(9):1577.; N Engl J Med 1995; 333:550.; Aust N Z J Med 1991; 21:714.; Nat Rev Endocrinol 2014; 10:143.].” it should be indicated that there is an absolute necessity to modify the information submitted in the sentence: “Such antidiabetic drugs can significantly reduce blood sugar levels, but most are not effective in treating insulin resistance. In addition, long-term use of many of these drugs can cause side effects, leading to liver and kidney damage [iv].
  3. Disadvantages and advantages of the animal model – used in the study – have not been convincingly discussed. It is need to be added;
  4. The choice of the animal model has not been justified. It is absolutely necessary – at least briefly – to discuss the issue in question;
  5. The choice of the active substance of natural origin – Silk sericin protein from Bombyx mori cocoons – has not been justified. It is absolutely necessary – at least briefly – to discuss the issue in question;
  6. It is absolutely necessary – at least briefly – discuss the most important interactions of the described active substances;
  7. In the case of using a control substance - metformin and the examined substance – Silk sericin protein – the Authors should compare the mechanism of the effects of the mentioned substances on not only carbohydrate metabolism but also on the liver tissue.
  8. Chapter 5 – Conclusion (in its present form) – represents the summary of the results but not the conclusions of the examinations implemented in the study protocol. Chapter 5 – Conclusion – absolutely needs to be written again.

Reviewer 3 Report

The article entitled “Repairing effect and mechanism of oral degraded sericin on liver injury in type II diabetic rats”  describes the dose dependent effect of sericin, a natural compound, on liver damages of diabetic rats. As positive control group the efficacy of metformin was evaluated. Even if the argument is relevant in order to find some molecules as alternative to metformin for the treatment of diabetic pathology, after reading the article I have found some limitations on the manuscript and on the research plan.

In Material and Methods section the preparation of Sericin peptides is not well described since it is not reported the concentration of sulfuric acid and  the centrifugation methodology.

In Results section it is reported:

“The co-efficient of the HLS group was significantly decreased compared with the model group”: please  clarify the statement: which group are you comparing the Sericin groups with the model or the normal group? It is not clear

“Treatment with sericin added to the food downregulated G6pase and PCK and upregu-lated GLK, PFK1, PKM2 (similar to the PC) in a dose–dependent manner”: The data does not support the hypothesis of the sericin effect on the enzymes respect to the Positive control

Impact on PI3K/Akt pathway: These data are reported for the same experimental conditions also in the article :

Exp Ther Med. 2018 Oct;16(4):3345-3352.

 doi: 10.3892/etm.2018.6615. Epub 2018 Aug 17.

Sericin enhances the insulin-PI3K/AKT signaling pathway in the liver of a type 2 diabetes rat model

Chengjun Song 1, Donghui Liu 1, Songhe Yang 1, Luyang Cheng 2, Enhong Xing 3, Zhihong Chen 1

Impact on AMPK/ACC pathway:

the effect on AMP kinase is not sustained by the data reported in figure 3;

Also it has no sense the inihibition of the expression of GLUT4 by low level of Sericin.

Impact on liver damage:

the results are not showed (the western blotting images) and are also not described in materials and methods section